# FINE-TUNING DIFFUSION POLICIES WITH BACKPROPAGATION THROUGH DIFFUSION TIMESTEPS

## ABSTRACT

Diffusion policies, widely adopted in decision-making scenarios such as robotics, gaming and autonomous driving, are capable of learning diverse skills from demonstration data due to their high representation power. However, the sub-optimal and limited coverage of demonstration data could lead to diffusion policies that generate sub-optimal trajectories and even catastrophic failures. While reinforcement learning (RL)-based fine-tuning has emerged as a promising solution to address these limitations, existing approaches struggle to effectively adapt Proximal Policy Optimization (PPO) to diffusion models. This challenge stems from the computational intractability of action likelihood estimation during the denoising process, which leads to complicated optimization objectives. In our experiments starting from randomly initialized policies, we find that online tuning of Diffusion Policies demonstrates much lower sample efficiency compared to directly applying PPO on MLP policies (MLP+PPO). To address these challenges, we introduce NCDPO, a novel framework that reformulates Diffusion Policy as a noise-conditioned deterministic policy. By treating each denoising step as a differentiable transformation conditioned on pre-sampled noise, NCDPO enables tractable likelihood evaluation and gradient backpropagation through all diffusion timesteps. This formulation enables direct optimization over the final denoised interactive actions without increasing MDP lengths. Our experiments demonstrate that NCDPO achieves sample efficiency comparable to MLP+PPO when training from scratch, outperforming existing methods in both sample efficiency and final performance across diverse benchmarks, including continuous robot control (with both state and vision inputs) and discrete multi-agent coordination tasks. Furthermore, our experimental results show that our method is robust to the number denoising timesteps.

## 1 INTRODUCTION

Recently, diffusion models have been widely adopted as policy classes in decision-making scenarios such as robotics (Chi et al.; Reuss et al., 2023; Ankile et al., 2024; Ze et al., 2024; Wang et al., 2024; Pearce et al., 2023), gaming (Pearce et al., 2023; Zhang et al., 2025), and autonomous driving (Liao et al., 2024; Yang et al., 2024). Although Diffusion Policies have shown remarkable capabilities in learning diverse behaviors from demonstration data (Chi et al.), Diffusion Policy could show sub-optimal performance when the demonstration data is sub-optimal or only covers a limited set of environment states. To further optimize the performance of pretrained policies, Reinforcement Learning (RL) is adopted as a natural choice for fine-tuning pre-trained Diffusion Policies through interaction with the environment.

Currently, the most effective approach, DPPO (*Diffusion Policy Policy Optimization*) (Ren et al., 2024) employs Policy Gradient (PG) approaches to enhance the performance of pre-trained Diffusion Policy in continuous control tasks. By treating the denoising process of Diffusion Policy as a low-level Markov Decision Process, DPPO optimizes the Gaussian likelihood of all denoising steps. However, through our extensive experiments, we find fine-tuning Diffusion Policies with RL faces a challenge of sample efficiency. Specifically, in our RL experiments starting from randomly initialized policies, we find that training Diffusion Policy with DPPO could lead to worse sample efficiency and final performance than training an MLP policy with standard RL. We hypothesize that the training efficiency gap occurs because DPPO uses a much longer MDP horizon for RL training, which impedes the sample efficiency of RL training. Therefore, a question becomes particularly important:

*Can we design a more effective fine-tuning approach for Diffusion Policy that avoids lengthening the MDP horizon during RL training?*

In this paper, we present *Noise-Conditioned Diffusion Policy Optimization* (NCDPO), a sample-efficienct reinforcement learning algorithm for fine-tuning Diffusion Policies *without increasing the MDP horizon*. Specifically, we first interpret the denoising process as a *noise-conditioned inference procedure*: we pre-sample the complete noise sequence for all diffusion steps and treat it as fixed, making the denoising trajectory deterministic and allowing exact backpropagation. This enables us to reformulate the RL objective to apply PPO only to *interactive actions*—the fully denoised actions used for environment interaction—and apply *Backpropagation through Diffusion Timesteps (BPDT)* to propagate the PPO gradients backward through the entire denoising chain.

We also conduct extensive experiments across diverse domains, including continuous robot control with both state and vision inputs, as well as discrete multi-agent coordination. In all settings, NCDPO consistently surpasses baselines in both sample efficiency and final performance. On vision-based manipulation tasks, it achieves up to $4\times$ higher sample efficiency than DPPO, highlighting its promise for real-world applications. Ablation studies further demonstrate that NCDPO is robust to the choice of diffusion timesteps. We attribute these improvements to the *shorter effective MDP length* and the *more accurate gradient estimation enabled by BPDT*

## 2 RELATED WORK

**Diffusion Models and Diffusion Policies.**    Diffusion-based generative models have demonstrated remarkable effectiveness in the domains of visual content generation (Rombach et al., 2022; Song et al., 2020; Ramesh et al., 2021).One central capability of Diffusion Models is the denoising process that iteratively refines sampled noises into clean datapoints (Ho et al., 2020; Sohl-Dickstein et al., 2015; Song & Ermon, 2019). Beyond their success in content generation, diffusion models have increasingly been adapted for decision-making tasks across a range of domains, including robotics (Chi et al.; Reuss et al., 2023; Ankile et al., 2024; Ze et al., 2024; Wang et al., 2024; Pearce et al., 2023), autonomous driving (Liao et al., 2024; Yang et al., 2024), and gaming (Pearce et al., 2023; Zhang et al., 2025). In robotics, most existing work trains Diffusion Policies through imitation learning. For instance, Reuss et al. (2023) predict future action chunks using goal-conditioned imitation learning, while (Ze et al., 2024) integrate Diffusion Policies with compact 3D representations extracted from point clouds. To further enhance the quality of generated behaviors, return signal or goal conditioning is applied to encourage the generation of high-value actions (Janner et al., 2022; Ajay et al., 2022; Liang et al., 2023).

**Fine-tuning Diffusion Policy with Reinforcement Learning.**    Recent works have aimed to enhance learned Diffusion Policy through fine-tuning with Reinforcement Learning approaches. A line of work has been focusing on integrating Diffusion Policies with Q-learning using offline data (Chen et al., 2022; Kang et al., 2023; Wang et al., 2022; Goo & Niekum, 2022; Rigter et al., 2023; Zhu et al., 2023; Psenka et al., 2023; Gao et al., 2025; Fang et al., 2024). In addition to offline reinforcement learning, recent advancements have explored fine-tuning Diffusion Policies with online RL algorithms, for example, aligning the score function with the action gradient (Yang et al., 2023), or employing the diffusion model as a policy extraction mechanism within implicit Q-learning (Hansen-Estruch et al., 2023). Most recently, Ren et al. (2024) formulates the denoising process of Diffusion Policy as a "Diffusion MDP", enabling the application of RL algorithms to optimize all denoising steps with online feedback. In this work, we investigate an alternative representation for the denoising process that enables sample efficient fine-tuning of Diffusion Policy.

## 3 PRELIMINARY

**Markov Decision Process.**  A Markov Decision Process (MDP) is defined as a tuple $\mathcal{M} = \langle \mathcal{S}, \mathcal{A}, P_0, P, R, \gamma \rangle$ where $\mathcal{S}$ denotes the state space, $\mathcal{A}$ is the action space, $P_0$ is the distribution of initial states, $P$ is the transition function, $R$ is the reward function and $\gamma$ is the discount factor. At timestep $t$, a policy $\pi$ generates an action $a_t \in \mathcal{A}$ at state $s_t$. The goal is to find a policy $\pi$ that

maximizes the objective of expected discounted return,

$$J(\pi) = \mathbb{E}_{s_t, a_t}\left[\sum_{t \geq 0} \gamma^t R(s_t, a_t)\right] \tag{1}$$

**Proximal Policy Optimization (PPO).** PPO is a reinforcement learning approach that optimizes the policy by estimating the policy gradient. In each iteration, given the last iteration policy $\pi_{\theta_k}$, PPO maximizes the clipped objective,

$$L(\theta|\theta_k) = \mathbb{E}_\tau\left[\sum_t \min\left(\frac{\pi_\theta(a_t|s_t)}{\pi_{\theta_k}(a_t|s_t)} A^{\pi_{\theta_k}}(s_t, a_t),\right.\right.$$
$$\left.\left. \text{clip}\left(\frac{\pi_\theta(a_t|s_t)}{\pi_{\theta_k}(a_t|s_t)}, 1 - \epsilon, 1 + \epsilon\right) A^{\pi_{\theta_k}}(s_t, a_t)\right)\right] \tag{2}$$

where $A^{\pi_{\theta_k}}(s_t, a_t)$ is the estimated advantage for action $a_t$ at state $s_t$.

**Diffusion Policy.** Diffusion Policy $\pi_\theta$ is a diffusion model that generates actions $a$ by conditioning on states $s$. In Diffusion Policy training, the *forward process* gradually adds Gaussian noise to the training data to obtain a chain of noisy datapoints $a^0, a^1, \ldots, a^K$,

$$q(a^{1:K}|a^0) := \prod_{k=1}^K q(a^k|a^{k-1}), \qquad q(a^k|a^{k-1}) := \mathcal{N}(a^k; \sqrt{1 - \beta_k}a^{k-1}, \beta_k I) \tag{3}$$

Diffusion Policy could generate actions with a *reverse process* or *denoising process* that gradually denoises a Gaussian noise $a^K \sim \mathcal{N}(a^K; 0, I)$ with learned Gaussian transitions,

$$\pi_\theta(a^{0:K}|s) := \prod_{k=1}^K \pi_\theta(a^{k-1}|a^k, s), \qquad \pi_\theta(a^{k-1}|a^k, s) := \mathcal{N}(a^{k-1}|\mu_\theta(a^k, k, s), \sigma_k^2 I) \tag{4}$$

where $\sigma$ is a fixed noise schedule for action generation, $\beta$ denotes the forward process variances and is held as constant, and $\theta$ is the parameter of Diffusion Policy. To avoid ambiguity, we use *interactive actions* to denote the action $a^0$ that is used for interacting with the environment and *latent actions* to denote actions $a^1, \cdots, a^K$ that are generated during the denoising process. For more training details on diffusion models, please refer to Ho et al. (2020).

**Diffusion Policy Policy Optimization (DPPO).** Note that the action likelihood $\pi_\theta(a_t^0|s_t)$ of Diffusion Policy $\pi_\theta$ is intractable,

$$\pi_\theta(a_t^0|s_t) = \int_{a_t^1, \cdots, a_t^K} \mathbb{P}[a_t^0, \cdots, a_t^K|s_t, \pi_\theta] \cdot da_t^1 \cdots da_t^K$$

The intractability of the action likelihood makes it impossible to directly fine-tune Diffusion Policy with PPO since the RL loss (Eq. 2) requires computing the exact action likelihood. To address this challenge, DPPO (Ren et al., 2024) proposes to formulate the denoising process as a low-level "Diffusion MDP" $\mathcal{M}_{\text{Diff}}$. In $\mathcal{M}_{\text{Diff}}$, a state is defined as a combination of the environment state and a *latent action* $\hat{s}_t^k = (s_t, a_t^k)$. For $k = K, \cdots, 1$, the transition from $\hat{s}_t^k$ to $\hat{s}_t^{k-1}$ represents the denoising process and takes no actual change on the environment state. For a denoising step $k \in [1, K]$, the state $\hat{s}_t^k = (s_t, a_t^k)$ transits to $\hat{s}_t^{k-1} = (s_t, a_t^{k-1})$. After the denoising process is finished at $k = 0$, the interactive action $a_t^0$ is used to interact with the environment and triggers the environment transition, i.e. the next state would be $\hat{s}_{t+1}^K = (s_{t+1}, a_{t+1}^K)$ where $s_{t+1} \sim P(s_t, a_t^0)$ and $a_{t+1}^K \sim N(0, I)$ is a newly generated Gaussian noise.

## 4 SAMPLE EFFICIENCY CHALLENGE FOR DIFFUSION POLICY FINE-TUNING

In this section, we study the sample efficiency of fine-tuning Diffusion Policies with reinforcement learning using one existing method. Concretely, we compare (i) training a diffusion-based policy with DPPO (denoted *DP+DPPO*) and (ii) training a standard MLP policy with PPO (denoted *MLP+PPO*).

To isolate the RL process, both policies are randomly initialized before RL and no behavior cloning is used.

Experiments on two OpenAI Gym locomotion tasks (Walker2D and HalfCheetah) show that, despite the stronger representational power of Diffusion Policies (Chi et al.), *DP+DPPO is substantially less sample efficient than MLP+PPO* and typically converges to a lower final performance (Fig. 12).

We hypothesize the reason is that, by employing a two-level MDP formulation, DPPO significantly lengthens the MDP horizons in RL training to contain both the interactive actions and latent actions. This lengthened MDP horizon then results in difficulty in proper credit assignment. This observation prompts us to *design a different method that could fine-tune Diffusion Policies without lengthening the MDP horizon*.

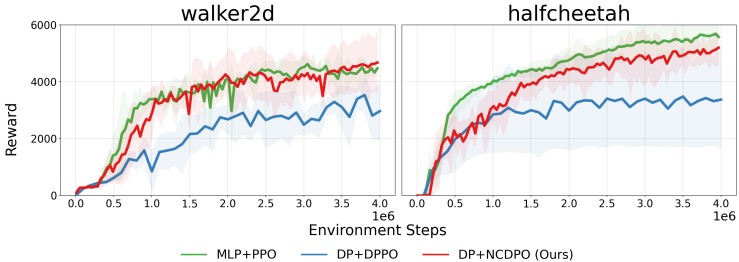

Figure 1: RL training from randomly initialized policy on Walker2D and HalfCheetah. Results are averaged over three seeds. Training curves indicate that DP+DPPO is less sample efficient than MLP+PPO and only achieves sub-optimal performance. Our approach, NCDPO, could fine-tune Diffusion Policy with high sample efficiency. Further study on the impact of MDP lengths can be found in Appendix. E

## 5 NOISE-CONDITIONED DIFFUSION POLICY OPTIMIZATION

As motivated in Sec. 4, treating the diffusion denoising process as part of the MDP can harm sample efficiency. In this section, we present a novel sample-efficient RL training method for Diffusion Policy, *Noise-Conditioned Diffusion Policy Optimization (NCDPO)*. NCDPO introduces backpropagation through diffusion timesteps by treating the diffusion denoising process as deterministic inference conditioned on pre-sampled noise, and applies PPO in the interactive action space. In Sec. 5.1 we formalize the noise-conditioned generative view. In Sec. 5.2 we show how PPO can be directly applied to the interactive actions instead of latent actions.

### 5.1 DENOISING PROCESS AS A NOISE-CONDITIONED INFERENCE PROCESS

**Noise-conditioned Action Generation.** Let $K$ be the number of denoising steps. A single denoising step can be written as

$$a^{k-1} = \mu_\theta(a^k, k, s) + \sigma_k z^k, \qquad z^k \sim \mathcal{N}(0, I), \tag{5}$$

for $k = K, K-1, \ldots, 1$, with $a^K \sim \mathcal{N}(0, I)$. The only stochastic terms are the Gaussian noises $a^K$ and $z^{1:K}$, and the map $\mu_\theta$ is deterministic given its inputs. Thus the sampling procedure naturally splits into a *noise sampling phase* and a *deterministic inference phase*.

In the noise sampling phase we draw

$$a^K \sim \mathcal{N}(0, I), \qquad z^k \sim \mathcal{N}(0, I) \text{ for } k = 1, \ldots, K. \tag{6}$$

In the deterministic inference phase we recursively apply Eq. (5) to obtain $a^{K-1}, \ldots, a^0$. Therefore the final action can be written as a deterministic function of the environment state and the sampled noises:

$$a^0 = \mu_\theta(\mu_\theta(\ldots \mu_\theta(a^K, K, s) \ldots, 2, s) + \sigma_2 \cdot z^2, 1, s) + \sigma_1 \cdot z^1$$
$$= f_\theta(s, a^K, z^{1:K}) \tag{7}$$

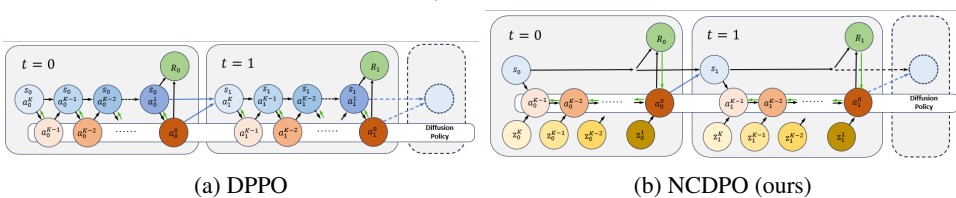

(a) DPPO               (b) NCDPO (ours)

Figure 2: DPPO adopts a two-layer MDP design by combining the environment state with latent actions to form augmented states. In contrast, in each step, NCDPO only applies PPO loss to the final interactive actions, resulting in shorter MDP lengths. (Eq. 10). Green arrows in the figure indicate gradient flow.

**MDP with Noise-augmented States.** As derived in Eq. 7, the denoising process can be divided into a noise sampling phase and a policy inference phase. We can incorporate the sampled noises into the MDP as part of the environment state.

Formally, for the original environment MDP $\mathcal{M} = \langle \mathcal{S}, \mathcal{A}, P_0, P, R, \gamma \rangle$ we introduce *MDP with noise-augmented states* $\mathcal{M}_{noise} = \langle \mathcal{S}_{noise}, \mathcal{A}, P_0, P_{noise}, R, \gamma \rangle$. In $\mathcal{M}_{noise}$, each state $s_{noise}$ consists of a environment state $s \in \mathcal{S}$ and Gaussian noises $a^K, z^1, \cdots, z^K$. $\mathcal{M}_{noise}$ shares the same action space and reward function as the original MDP $\mathcal{M}$. In each decision-making step, the environment state transits to a new one, and the noises are all re-sampled.

**Stochastic Noise-Conditioned Policy.** Given a noise-augmented state $s_{noise} = (s, a^K, z^{1:K})$, the deterministic inference phase of the denoising process can be represented as a *Noise-Conditioned Policy* $\pi_\theta^{NC}$ that generates the action $a^0$ using Eq. 7.

This noise-conditoned policy is a deterministic policy and could not be directly trained with PPO since the policy loss (Eq. 2) relies on a stochastic policy. Therefore, we introduce an additional operation to transform this deterministic policy into a stochastic one.

For continuous actions we use a Gaussian policy centered at $f_\theta$:

$$\pi_\theta^{\text{NC}}(\cdot \mid s, a^K, z^{1:K}) = \mathcal{N}\big(f_\theta(s, a^K, z^{1:K}), \ \text{diag}(\sigma_{\text{act}})^2\big), \tag{8}$$

where $\sigma_{\text{act}}$ is a learnable vector of standard deviations. For discrete actions we treat $f_\theta(\cdot)$ as logits and apply a temperatureed softmax:

$$\pi_\theta^{\text{NC}}(a^0 = i \mid s, a^K, z^{1:K}) \propto \exp\big(f_\theta(s, a^K, z^{1:K})_i / T\big). \tag{9}$$

With this construction, PPO can optimize over the interactive action $a^0$ directly while the diffusion model deterministically generates the stochastic policy conditioned on pre-sampled noise.

## 5.2 FINE-TUNING THE NOISE-CONDITIONED POLICY WITH PPO

Under the formulation of NCDPO, at each environment timestep $t$ we first sample Gaussian noise $a_t^K, z_t^{1:K}$ and then generate $a_t^0 = f_\theta(s_t, a_t^K, z_t^{1:K})$ , and then sample the executed action from $\pi_\theta^{\text{NC}}$ as in Eq. (8) or (9). We store the tuple

$$(s_t, \ a_t^K, \ z_t^{1:K}, \ a_t^0, \ r_t, \ s_{t+1})$$

in the experience buffer. During optimization we reuse the stored noises $a_t^K, z_t^{1:K}$ to deterministically recompute $a_t^0$ and compute the PPO loss on those actions; gradients are backpropagated through the entire denoising computation $f_\theta$. This design lets PPO operate on environment actions while updating all denoising steps end-to-end.

We instantiate the PPO objective in the standard clipped form, optimizing the probability of interactive actions produced by the noise-conditioned policy:

$$L(\theta \mid \theta_k) = \mathbb{E}_{a_t^0 \sim \pi_{\theta_k}^{\mathrm{NC}}(\cdot \mid s_t, a_t^K, z_t^{1:K})} \left[ \sum_t \min\left( \frac{\pi_\theta^{\mathrm{NC}}(a_t^0 \mid s_t, a_t^K, z_t^{1:K})}{\pi_{\theta_k}^{\mathrm{NC}}(a_t^0 \mid s_t, a_t^K, z_t^{1:K})} A_t, \right. \right.$$
$$\left. \left. \mathrm{clip}\left( \frac{\pi_\theta^{\mathrm{NC}}(a_t^0 \mid s_t, a_t^K, z_t^{1:K})}{\pi_{\theta_k}^{\mathrm{NC}}(a_t^0 \mid s_t, a_t^K, z_t^{1:K})}, 1 - \epsilon, 1 + \epsilon \right) A_t \right) \right], \tag{10}$$

**Rollout / training pseudocode.** For clarity, we provide a short pseudocode of the rollout and training loop used by NCDPO.

As illustrated in Algo 1, in policy rollout process, each step begins by sampling a sequence of noises, which are then used by the Diffusion Policy to generate the corresponding action. These sampled noises are stored in the buffer. During training phase, the stored noises are reused to recompute the actions, enabling gradient backpropagation through the entire denoising process. This allows PPO to directly update all denoising steps of the diffusion policy.

---

**Algorithm 1** NCDPO

---

**Require:** Noise-conditionoed policy $\pi_\theta^{NC}$, noise scheduler $\sigma$
1: **Parameters:** $\gamma \in [0, 1), \varepsilon \in (0, 1), N_{\mathrm{episodes}}, N_{\mathrm{PPO}}$
2: **for** $e = 1, 2, \dots, N_{\mathrm{episodes}}$ **do**
3:     buffer $\leftarrow \emptyset$
4:     **for** $t = 0, 1, 2, \dots, T - 1$ **do**
5:         $a_t^K, z_t^{1:K} \sim \mathcal{N}(0, I)$
6:         Sample $a_t^0$ from $\pi^{NC}(\cdot | z_t^{1:K}, a_t^K, s_t)$
7:         $\log \pi_t^{NC} \leftarrow \pi^{NC}(a_t^0 | z_t^{1:K}, a_t^K, s_t)$
8:         Execute $a_t$, observe $r_t, s_{t+1}$
9:         buffer $\leftarrow$ buffer $\cup \{s_t, a_t^K, r_t, \log \pi_t, z_t^{1:K}\}$
10:     **end for**
11:     **for** epoch $= 1, 2, \dots, N_{\mathrm{PPO}}$ **do**
12:         **for** mini-batch $b = 1, 2, \dots$ **do**
13:             Calculate PPO loss $L(\theta | \theta_k)$ in Eq. 10, backpropagate gradients through diffusion timesteps and update parameter $\theta$
14:         **end for**
15:     **end for**
16: **end for**

---

As Fig.2 shows, NCDPO models the denoising process as deterministic generation conditioned on pre-sampled noise $z_t^{1:K}$. During inference, interactive actions are obtained through recursive model inference in Eq. 7 and applying the action sampling step in Eq. 8 and Eq. 9.

### 5.3 NCDPO IS UNBIASED

In this subsection, we provide a formal proof that the gradient estimate of NCDPO is an unbiased estimator of the true objective gradient. We show that applying PPO to the final interactive action $a^0$, while backpropagating through the deterministic denoising chain $f_\theta$, is mathematically equivalent to optimizing the expected reward over the induced distribution. The high-level idea of the proof is that NCDPO applies reparametrization trick similar to that used in methods like VAE. *This gives NCDPO the unique advantage of being able to optimize the diffusion policy as long as the final policy is parameterized by $a^0$* (for example, Gaussian in continuous space or Softmax in discrete case).

**Preliminaries.** Let $\xi = (a^K, z^{1:K})$ denote the sequence of sampled Gaussian noises used in the denoising process. Since each noise term is drawn from $\mathcal{N}(0, I)$, $\mathbb{E}_\xi$ represents the expectation over the joint distribution of these independent Gaussian variables.

Recall from Eq. (7) that the deterministic inference phase produces a mean action $f_\theta(s, \xi)$. The final executed action $a^0$ is sampled from the noise-conditioned policy:

$$a^0 \sim \pi_\theta^{\mathrm{NC}}(\cdot \mid s, \xi). \tag{11}$$

For the continuous case defined in Eq. (8), this is a Gaussian $\mathcal{N}(f_\theta(s, \xi), \sigma_{\mathrm{act}}^2)$.

The true objective $J(\theta)$ is the expected reward over both the pre-sampled noise $\xi$ and the final action sampling noise:

$$J(\theta) = \mathbb{E}_\xi \left[ \mathbb{E}_{a^0 \sim \pi_\theta^{\mathrm{NC}}(\cdot|s,\xi)}[R(a^0)] \right]. \tag{12}$$

**NCDPO Gradient.** NCDPO applies the policy gradient theorem to the noise-conditioned policy $\pi_\theta^{\mathrm{NC}}$. The gradient of the NCDPO objective with respect to $\theta$ is:

$$\nabla_\theta J_{\mathrm{NCDPO}} = \mathbb{E}_\xi \left[ \mathbb{E}_{a^0 \sim \pi_\theta^{\mathrm{NC}}} \left[ \nabla_\theta \log \pi_\theta^{\mathrm{NC}}(a^0 \mid s, \xi) \, R(a^0) \right] \right]. \tag{13}$$

By applying the chain rule, we can expand the gradient of the log-probability term:

$$\nabla_\theta \log \pi_\theta^{\mathrm{NC}}(a^0 \mid s, \xi) = \nabla_{f_\theta} \log \pi_\theta^{\mathrm{NC}}(a^0 \mid s, \xi) \cdot \nabla_\theta f_\theta(s, \xi). \tag{14}$$

Substituting this into Eq. (13), we obtain the expanded gradient form:

$$\nabla_\theta J_{\mathrm{NCDPO}} = \mathbb{E}_\xi \left[ \left( \mathbb{E}_{a^0} \left[ \nabla_{f_\theta} \log \pi_\theta^{\mathrm{NC}}(a^0 \mid s, \xi) \cdot R(a^0) \right] \right) \nabla_\theta f_\theta(s, \xi) \right]. \tag{15}$$

**Equivalence to True Gradient.** We compare this to the gradient of the true objective. Using the standard score function identity on the inner expectation over $a^0$, we know that for a fixed $\xi$:

$$\mathbb{E}_{a^0 \sim \pi_\theta^{\mathrm{NC}}} \left[ \nabla_{f_\theta} \log \pi_\theta^{\mathrm{NC}}(a^0 \mid s, \xi) \, R(a^0) \right] = \nabla_{f_\theta} \mathbb{E}_{a^0 \sim \pi_\theta^{\mathrm{NC}}}[R(a^0)]. \tag{16}$$

This equality holds because $\pi_\theta^{\mathrm{NC}}$ is a distribution parameterized by $f_\theta$. Substituting Eq. (16) back into Eq. (15):

$$\begin{aligned}
\nabla_\theta J_{\mathrm{NCDPO}} &= \mathbb{E}_\xi \left[ \nabla_{f_\theta} \mathbb{E}_{a^0}[R(a^0)] \cdot \nabla_\theta f_\theta(s, \xi) \right] \\
&= \mathbb{E}_\xi \left[ \nabla_\theta \mathbb{E}_{a^0 \sim \pi_\theta^{\mathrm{NC}}(\cdot|s,\xi)}[R(a^0)] \right] \\
&= \nabla_\theta \mathbb{E}_{\xi, a^0}[R(a^0)] = \nabla_\theta J(\theta).
\end{aligned} \tag{17}$$

Thus, the gradient estimated by NCDPO is equivalent to the gradient of the total expected reward with respect to $\theta$.

# 6 EXPERIMENTS

In this section, we provide a comprehensive evaluation of NCDPO across a variety of challenging environments. We first describe the experimental setup in Sec. 6.1, then present results on continuous robot control tasks (Sec. 6.2) and discrete multi-agent coordination tasks (Sec. 6.3). Finally, we report ablation studies in Sec. 6.4 to assess the robustness of NCDPO. Wall-clock results for the main experiments are reported in Appendix. F.

Through extensive experiments aimed to evaluate the capability of NCDPO across diverse continuous control scenarios, our main observations are summarized as follows:

- *NCDPO delivers the best performance:* Across all scenarios, NCDPO achieves the best overall performance. Compared with DPPO, it yields an average reward improvement of $30.6\%$ on OpenAI Gym locomotion tasks, $15.4\%$ on Franka Kitchen tasks, and an average success rate improvement of $68.9\%$ on Robomimic tasks with visual inputs (using 4M samples for the transport-vision task).

- *NCDPO achieves the highest sample efficiency.* NCDPO consistently attains the highest sample efficiency. Relative to DPPO, it improves sample efficiency by $800.0\%$ in OpenAI Gym locomotion tasks, $34.6\%$ in Franka Kitchen tasks, $97.3\%$ in Robomimic tasks with state inputs, and $313.8\%$ in Robomimic tasks with visual inputs on average. Here, we compare the number of samples required to reach the peak performance achieved by DPPO.

- *NCDPO is robust to the number of denoising steps.* NCDPO consistently maintains strong performance and high sample efficiency even when fine-tuning Diffusion Policies with increased numbers of denoising steps.

- *Q-learning methods underperforms in long-horizon, sparse-reward tasks:* Q–learning methods underperform and exhibits extreme instability on long-horizon, sparse-reward tasks such as Franka Kitchen and Robomimic, likely due to the increased difficulty of accurately estimating Q-values in these settings. This underscores the importance of on-policy, PPO-style algorithms for tackling such challenging tasks.

For continuous robot control tasks, we compare **NCDPO** to several baselines: standard Gaussian policies with **MLP** parameterization; **DPPO** (Ren et al., 2024); **DRWR** and **DAWR** (Ren et al., 2024), which build on reward-weighted regression (Peters & Schaal, 2007) and advantage-weighted regression (Peng et al., 2019), respectively; **DIPO** (Yang et al., 2023), which uses action gradients as the score function during denoising; and Q-learning–based methods such as **IDQL** (Hansen-Estruch et al., 2023) and **DQL** (Wang et al., 2022). To further demonstrate the advantages of our PPO-based approach over Q–learning methods, we also compare against **DAC** (Fang et al., 2024) and **BDPO** (Gao et al., 2025) on sparse-reward tasks. To ensure a fair comparison, these methods were adapted to an offline-to-online setting by adding a replay buffer.

For the discrete multi-agent environment, Google Research Football, we compare NCDPO with MLP policies trained with Multi-Agent Proximal Policy Optimization (MAPPO) (Yu et al., 2022). The MAPPO baseline is initialized via behavior cloning using a cross-entropy loss.

## 6.1 Environmental Setup

**OpenAI Gym locomotion.** We evaluate on standard locomotion benchmarks from OpenAI Gym (Brockman, 2016): `Hopper-v2`, `Walker2D-v2`, and `HalfCheetah-v2`. Pre-trained Diffusion Policies were trained on the D4RL "medium" dataset (Fu et al., 2020), which contains diverse pre-recorded trajectories. Fine-tuning is performed with **dense** rewards.

**Franka Kitchen.** We evaluate on the Franka Kitchen benchmarks (Gupta et al., 2019), where agents must complete four ordered subtasks. Pretraining datasets are `complete`, `mixed`, and `partial` from D4RL (Fu et al., 2020). Fine-tuning uses **sparse** rewards (reward = 1 when a subtask is completed).

**Robomimic.** We evaluate on Robomimic manipulation tasks (Mandlekar et al., 2021): `Lift`, `Can`, `Square`, and `Transport`. To ensure temporal consistency, we apply action chunking with chunk size 4 for `Lift`, `Can`, and `Square`, and chunk size 8 for `Transport`, following (Ren et al., 2024). All Robomimic tasks are fine-tuned with **sparse** success/failure rewards.

**Google Research Football.** To evaluate performance in large discrete joint-action spaces, we test three multi-agent scenarios: `3 vs 1 with Keeper`, `Counterattack Hard`, and `Corner`. We adopt a centralized control strategy: a single Diffusion Policy generates joint actions for all agents simultaneously. Base Diffusion Policies are pre-trained to output one-hot vectors representing ground-truth actions. Because no public dataset exists for these scenarios, we construct a pretraining dataset by training multiple MAPPO agents (Yu et al., 2022) with different random seeds and early-stopping schedules; this produces a diverse set of trajectories with varying success rates and tactics.

## 6.2 Evaluation on Continuous Robot Control Tasks

**State Inputs.** Experimental results in Fig. 13 and Table 1 demonstrate that NCDPO consistently achieves the strongest performance, highest robustness, and superior sample efficiency across all evaluated tasks with state input. These findings highlight substantial improvements over DPPO and other baselines.

Notably, the `Transport` scenario is exceptionally challenging: it requires coordinating two robot arms across multiple complex sub-tasks, with extremely long horizons and sparse rewards. To the best of our knowledge, no reinforcement learning algorithm other than DPPO—whether off-policy or on-policy—has achieved a success rate above 50% on this task. The strong performance of NCDPO in this setting highlights its robustness and broad applicability.

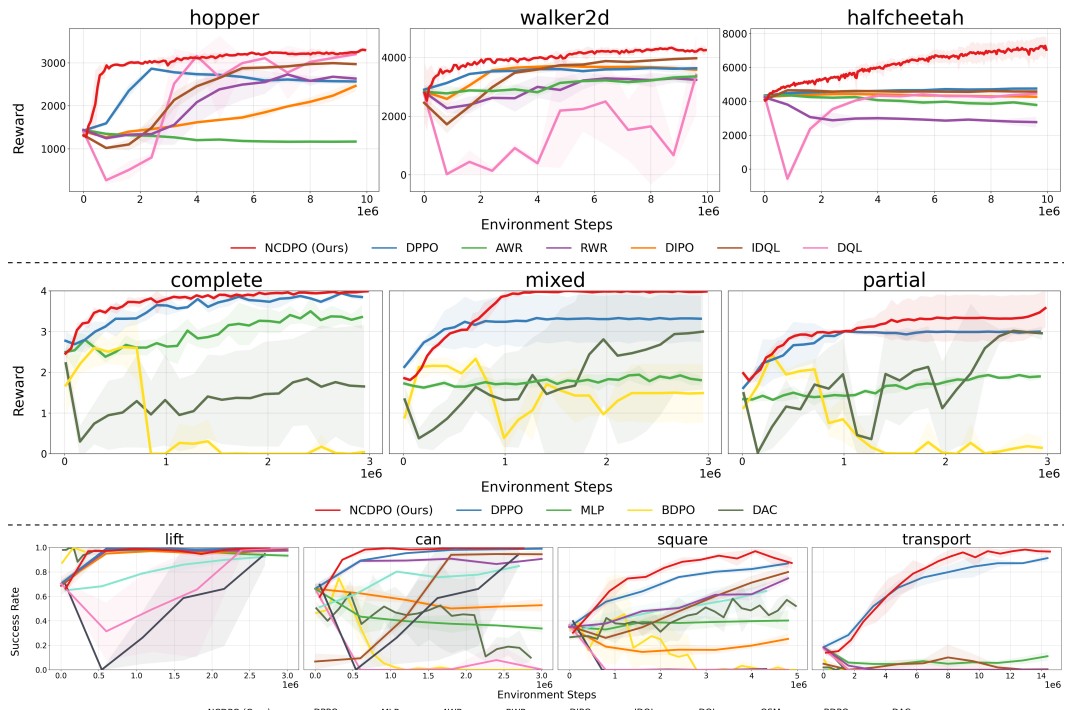

Figure 3: Performance comparisons on OpenAI Gym locomotion, Franka Kitchen, and Robomimic tasks. Results are averaged over three seeds. NCDPO (ours) achieves the strongest performance.

**Vision Inputs.** We also evaluate with *vision inputs* on the two most challenging Robomimic tasks (`Square` and `Transport`). Results are demonstrated in Fig. 4. Remarkably, NCDPO achieves substantial performance gains and attains roughly three times higher sample efficiency than DPPO on these visual tasks, underscoring its surprising effectiveness and strong potential for real-world applications.

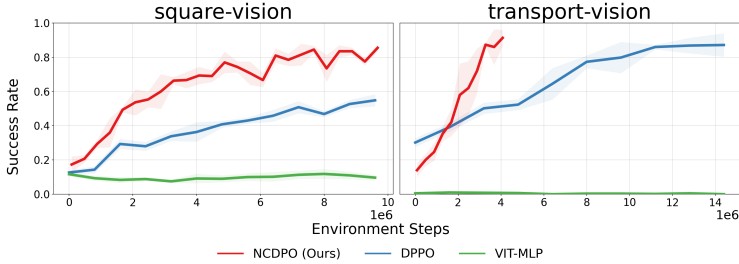

Figure 4: Performance comparison on Robomimic visual tasks. Results are averaged over three seeds. NCDPO (ours) achieves the strongest performance. VIT-MLP uses a Vision Transformer encoder and a Gaussian policy parameterize by MLP.

### 6.3 EVALUATION ON DISCRETE MULTI-AGENT COORDINATION TASKS

We next evaluate NCDPO on Google Research Football, a cooperative multi-agent benchmark. As shown in Fig. 5 and Table 2, NCDPO consistently outperforms the MAPPO (MLP) baseline across all three scenarios. These results highlight the strength of Diffusion Policies in modeling complex and diverse demonstration data, as well as the effectiveness of the DP+NCDPO fine-tuning procedure during fine-tuning phase. Moreover, they demonstrate the versatility of NCDPO, showing its effectiveness not only lies in continuous robot control tasks but also in environments with large discrete action spaces.

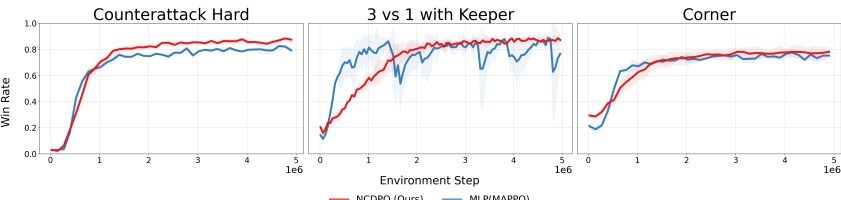

Figure 5: Performance comparison in Google Research Football. Results are averaged over at least three seeds. NCDPO (ours) exhibits strong performance and stability.

### 6.4 NCDPO IS ROBUST TO THE NUMBER OF DENOISING STEPS

We study the effect of varying the number of denoising steps in the diffusion model. Results in Fig. 6 indicate that NCDPO is robust to this choice across tasks. We hypothesize that this robustness stems from how gradients propagate through time in the diffusion process, which yields more accurate gradient estimates during fine-tuning.

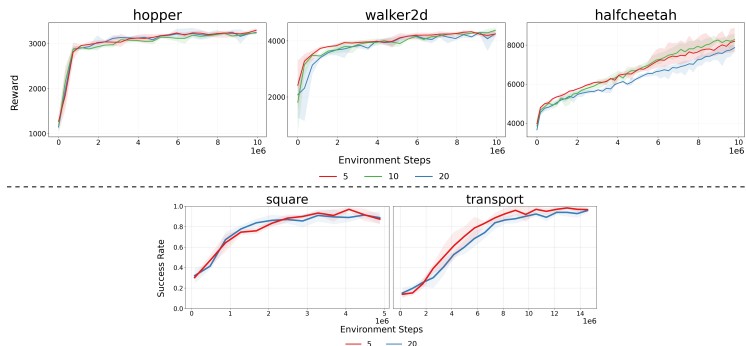

Figure 6: Ablation study on denoising steps in OpenAI Gym locomotion tasks and Robomimic tasks.

## 7 CONCLUSION AND LIMITATIONS

We present NCDPO, a novel approach for fine-tuning Diffusion Policies through Proximal Policy Optimization that exhibits strong performance across continuous and discrete control domains. Our key innovation lies in backpropagating gradients through diffusion timesteps by reformulating the diffusion denoising process as a noise-conditioned stochastic policy. Through extensive experiments across locomotion, manipulation, and multi-agent cooperation scenarios, we demonstrate that NCDPO achieves superior sample efficiency and final performance compared to existing diffusion RL approaches. NCDPO's ability to handle both continuous and discrete action spaces suggests its potential as a general-purpose policy optimization framework.

Our study focuses on the algorithmic development and evaluation of NCDPO in simulated settings. Consequently, we have not yet explored sim-to-real transfer on physical robots. These choices reflect our emphasis on fine-tuning methodology. Extending NCDPO to real-world deployment remains to be implemented in future work.

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

## A  SELF-IMITATION REGULARIZER

When directly fine-tuning Diffusion Policies using policy gradient methods, we observe a structure collapse issue—namely, the Diffusion Policy fails to maintain consistency between the forward and reverse processes. To preserve the structural integrity of the diffusion model, we introduce self-imitation regularization. Specifically, we perform behavior cloning on the trajectories generated in the previous episode. Empirically, we find that this regularization significantly reduces the behavior cloning loss. In contrast, without it, this behavior cloning loss will keep increasing, indicating structural degradation in the Diffusion Policy.

## B  ADDITIONAL EXPERIMENTAL RESULTS

### B.1  SCORES FOR CONTINUOUS ROBOT CONTROL TASKS

| Scenario | NCDPO (Ours) | DPPO | AWR | IDQL | DQL | RWR | DIPO | DAC | BDPO | QSM | MLP |
|---|---|---|---|---|---|---|---|---|---|---|---|
| hopper | **3297.3 (47.8)** | 2566.6 (51.1) | 1168.9 (30.5) | 2970.2 (5.2) | 3200.7 (30.1) | 2633.6 (94.0) | 2463.1 (127.6) | – | – | – | – |
| hopper-replay | **3345.6 (71.9)** | 2989.0 (86.9) | 2142.3 (183.8) | 3076.9 (29.8) | 3159.7 (94.8) | 2718.3 (79.9) | 2834.1 (25.0) | – | – | – | – |
| hopper-expert | **3528.7 (51.9)** | 2672.6 (135.1) | 1062.5 (13.1) | 3440.5 (10.8) | 3153.7 (733.4) | 2964.1 (121.3) | 3327.0 (32.7) | – | – | – | – |
| walker2d | **4248.8 (137.6)** | 3632.1 (55.9) | 3353.9 (296.9) | 3972.8 (17.3) | 3405.2 (1322.7) | 3238.3 (116.8) | 3581.6 (70.3) | – | – | – | – |
| walker2d-replay | **4544.6 (163.0)** | 3770.5 (154.5) | 2719.0 (83.8) | 4373.9 (91.7) | 3846.3 (1358.0) | 2483.0 (215.7) | 3180.8 (396.3) | – | – | – | – |
| walker2d-expert | **5060.9 (82.9)** | 4935.6 (73.3) | 4458.7 (195.3) | 4863.9 (95.3) | 2416.7 (1708.3) | 3831.7 (243.8) | 4980.0 (50.2) | – | – | – | – |
| halfcheetah | **7058.8 (635.2)** | 4758.3 (41.8) | 3788.7 (166.5) | 4584.4 (45.3) | 4459.1 (309.5) | 2773.1 (310.0) | 4272.4 (33.2) | – | – | – | – |
| halfcheetah-replay | **7000.1 (113.9)** | 4181.6 (24.6) | 3501.1 (40.3) | 4295.8 (53.0) | 4223.1 (177.3) | 1776.0 (101.3) | 3362.5 (49.5) | – | – | – | – |
| halfcheetah-expert | **8079.3 (392.2)** | 4663.2 (92.6) | 3723.2 (131.4) | 4499.4 (57.6) | 4172.0 (465.3) | 2218.4 (168.4) | 4127.0 (12.1) | – | – | – | – |
| | | | | | | | | | | | |
| kitchen-complete-v0 | **4.0 (0.0)** | 3.8 (0.1) | – | – | – | – | – | 1.6 (0.4) | 0.0 (0.0) | – | 3.4 (0.3) |
| kitchen-mixed-v0 | **4.0 (0.0)** | 3.3 (0.6) | – | – | – | – | – | 1.7 (0.7) | 2.2 (0.2) | – | 1.8 (0.1) |
| kitchen-partial-v0 | **3.6 (0.5)** | 3.0 (0.0) | – | – | – | – | – | 1.1 (1.0) | 2.5 (1.1) | – | 1.9 (0.1) |
| | | | | | | | | | | | |
| lift | **100.0 (0.0)** | 99.7 (0.3) | 93.3 (1.8) | 99.2 (1.0) | 99.8 (0.3) | 97.5 (0.5) | 97.3 (0.8) | 97.2 (3.3) | **100.0 (0.0)** | 94.7 (9.2) | 92.5 (0.0) |
| can | **99.3 (1.2)** | 99.0 (1.0) | 33.8 (3.2) | 94.5 (3.1) | 0.3 (0.6) | 90.7 (0.8) | 52.8 (5.1) | 12.0 (2.8) | 0.0 (0.0) | 94.7 (9.2) | 84.8 (2.5) |
| square | **87.3 (4.5)** | 87.0 (2.3) | 40.3 (8.5) | 80.0 (5.0) | 0.0 (0.0) | 74.8 (2.6) | 25.3 (4.5) | 52.2 (3.9) | 0.0 (0.0) | 0.7 (0.6) | 64.5 (9.5) |
| transport | **96.7 (2.3)** | 91.3 (2.9) | 11.2 (3.5) | 0.5 (0.9) | 0.0 (0.0) | 0.0 (0.0) | 0.2 (0.3) | 0.0 (0.0) | 0.0 (0.0) | 0.0 (0.0) | 0.0 (0.0) |
| square-vision | **85.5 (2.1)** | 54.8 (3.5) | – | – | – | – | – | – | – | – | 9.7 (1.6) |
| transport-vision (4M) | **91.3 (4.6)** | 50.2 (3.3) | – | – | – | – | – | – | – | – | 0.8 (0.6) |

Table 1: Mean and standard deviation of performance across robot control tasks.

| Scenario | NCDPO (Ours) | MAPPO |
|---|---|---|
| 3 vs 1 with Keeper | **87.4**(2.7) | 75.1(12.3) |
| Counterattack Hard | **87.0**(2.5) | 80.0(2.0) |
| Corner | **78.3**(4.5) | 74.9(3.0) |

Table 2: Average evaluation success rate and standard deviation (over three seeds) on Google Research Football scenarios. The base Diffusion Policy and MLP policy are pre-trained on the same dataset. MLP policy is trained using Cross-Entropy loss.

### B.2  ABLATION STUDY

We observe that setting the action chunk size to one significantly improves performance in Gym environments. We hypothesize that this is due to the nature of these tasks, where agents must respond promptly to rapid and continuous changes in the environment. Smaller chunk sizes allow the policy to adapt its actions more frequently, which is crucial for achieving fine-grained control. The corresponding results are presented in Figure 7.

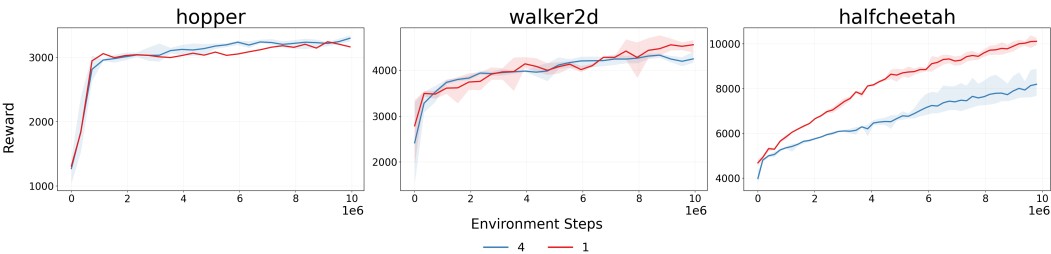

Figure 7: Ablation study on action chunk size in OpenAI Gym locomotion tasks.

additionally, in discrete action environments, modifying the noise scheduler to increase Gaussian noise during the denoising steps improves exploration without degrading overall performance, as shown in Figure 8. In discrete settings, the absolute values of the logits are less important than their relative magnitudes, which allows increased noise to encourage exploration while preserving policy effectiveness. To achieve this, we adjust the noise scheduler using parameters $\eta$ and $\beta_{\text{base}}$, increasing the noise level via the transformation:

$$\beta'_k = \beta_{base} \left( \frac{\beta_k}{\beta_{base}} \right)^\eta$$

where $\beta_k$ corresponds to the original noise schedule defined in Equation 3. In our implementation, we set $\beta_{base} = 0.7$.

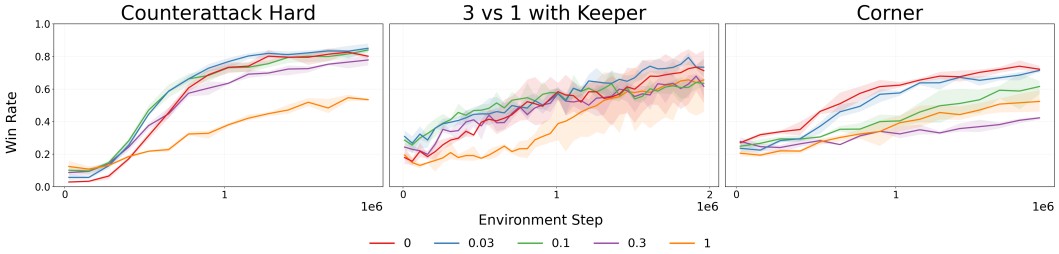

Figure 8: Ablation study on different values of $\eta$ in Google Research Football.

We further find that increasing the initial noise scale $\sigma_a$ in the acting layer enhances exploration. An ablation study conducted on Robomimic supports this finding:

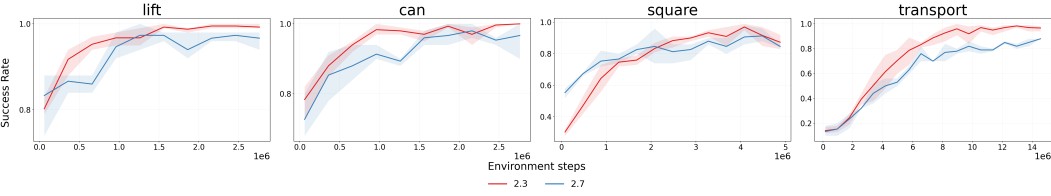

Figure 9: Ablation study of different choices of initial $\log \sigma_a$ in Robomimic tasks.

## B.3 FURTHER EXPERIMENTS IN OPENAI GYM LOCOMOTION TASKS.

We evaluated different training methods using datasets of varying quality for pretraining the base policy. The "medium-replay" dataset consists of replay buffer samples collected before early stopping, while the "medium-expert" dataset contains equal proportions of expert demonstrations and suboptimal rollouts Fu et al. (2020).

Regardless of dataset quality, NCDPO consistently outperforms all baselines, as shown in Figures 10 and 11.

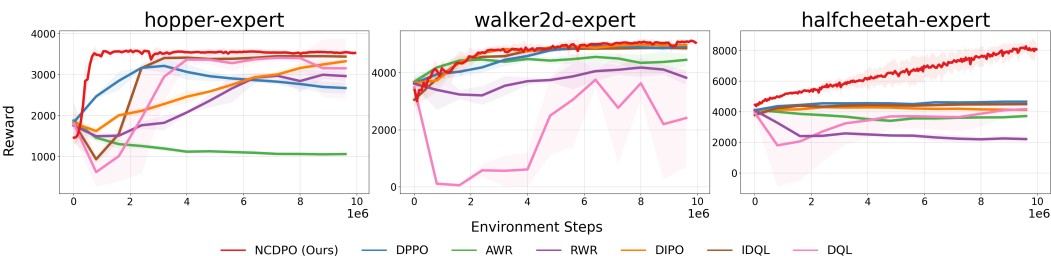

Figure 10: Pretraining with expert datasets.

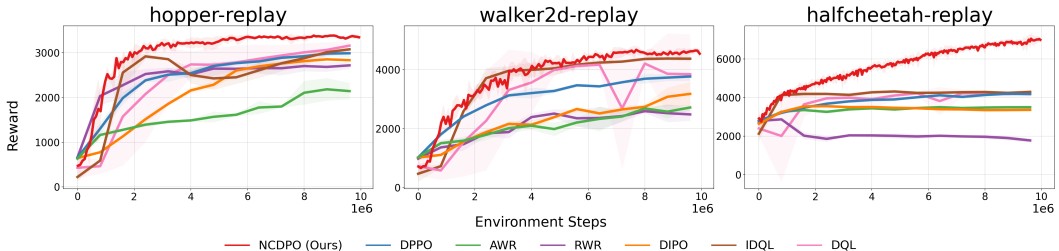

Figure 11: Pretraining with replay datasets.

### B.4 GOOGLE RESEARCH FOOTBALL DATA CURATION

For each scenario, we collected 200K environment steps per model. The win rates of the agents used for dataset generation are summarized in Table 3.

| Scenario | Win Rates |
|---|---|
| 3 vs 1 with Keeper | 0.93, 0.90, 0.70, 0.55 |
| Corner | 0.76, 0.75, 0.50, 0.50, 0.41 |
| Counterattack Hard | 0.90, 0.78, 0.70, 0.61, 0.56, 0.56 |

Table 3: Win rates of trained agents used for dataset collection in Google Research Football. Each model contributes 200,000 steps.

## C IMPLEMENTATION DETAILS AND HYPERPARAMETERS

For NCDPO, we apply Adam optimizer for actor and AdamW optimizer for critic. For all other baselines, AdamW optimizer is adopted.

For fair comparison, we adopt the same network architecture as DPPO Ren et al. (2024) and directly utilize their implementation for model structure. Our overall training framework is built upon a modified codebase of MAPPO Yu et al. (2022).

| Task | $\gamma$ | $\lambda$ | Action Chunk | Actor LR | Critic LR | Actor MLP Size | Critic MLP Size | Actor MLP Layers | $\eta$ | Initial Noise Log Std | $1/T$ | Denoising Step | Clone Epochs | Clone LR | Episode Length | Mini-batch Number | Environment Max Steps | Parallel Environments |
|---|---|---|---|---|---|---|---|---|---|---|---|---|---|---|---|---|---|---|
| Hopper | 0.995 | 0.985 | 4 | 3e-5 | 1e-3 | 1024 | 256 | 7 | 0 | -2 | - | 5 | 8 | 1e-3 | 256 | 1 | 1000 | 32 |
| Walker2d | 0.995 | 0.985 | 4 | 3e-5 | 1e-3 | 1024 | 256 | 7 | 0 | -2 | - | 5 | 8 | 1e-3 | 500 | 1 | 1000 | 32 |
| HalfCheetah | 0.99 | 0.985 | 4 | 3e-5 | 1e-3 | 1024 | 256 | 7 | 0 | -2 | - | 5 | 3 | 1e-4 | 500 | 1 | 1000 | 32 |
| lift | 0.999 | 0.99 | 4 | 3e-5 | 1e-3 | 1024 | 256 | 7 | 0 | -2.3 | - | 5 | 2 | 1e-3 | 300 | 4 | 300 | 200 |
| can | 0.999 | 0.99 | 4 | 3e-5 | 1e-3 | 1024 | 256 | 7 | 0 | -2.3 | - | 5 | 60 | 3e-4 | 300 | 4 | 300 | 200 |
| square | 0.999 | 0.99 | 4 | 3e-5 | 1e-3 | 1024 | 256 | 7 | 0 | -2.3 | - | 5 | 200 | 2e-4 | 400 | 4 | 400 | 200 |
| square-vision | 0.999 | 0.99 | 4 | 5e-4 | 1e-3 | 1024 | 256 | 7 | 0 | -2.3 | - | 5 | 200 | 2e-4 | 400 | 4 | 400 | 200 |
| transport | 0.999 | 0.99 | 8 | 3e-5 | 1e-3 | 1024 | 256 | 7 | 0 | -2.3 | - | 5 | 321 | 8e-4 | 800 | 4 | 800 | 200 |
| transport-vision | 0.999 | 0.99 | 4 | 5e-5 | 1e-3 | 768 | 256 | 3 | 0 | -2.3 | - | 5 | 300 | 1e-4 - 1e-5 | 800 | 4 | 800 | 100 |
| 3 vs 1 with Keeper | 0.99 | 0.95 | 1 | 3e-5 | 1e-3 | 1024 | 256 | 7 | 0.03 | - | 20 | 5 | 10 | 1e-3 | 200 | 1 | - | 50 |
| Corner | 0.99 | 0.95 | 1 | 3e-5 | 1e-3 | 1024 | 256 | 7 | 0.03 | - | 20 | 5 | 10 | 1e-3 | 500 | 1 | - | 50 |
| Counterattack Hard | 0.99 | 0.95 | 1 | 3e-5 | 1e-3 | 1024 | 256 | 7 | 0.03 | - | 20 | 5 | 10 | 1e-3 | 500 | 1 | - | 50 |

Table 4: Hyperparameter settings for different tasks of NCDPO.

| Task | $\gamma$ | $\lambda$ | Action Chunk | Actor LR | Critic LR | Actor MLP Size | Critic MLP Size | Actor MLP Layers | Denoising Step | Episode Length | Mini-batch Size | Environment Max Steps | Parallel Environments |
|---|---|---|---|---|---|---|---|---|---|---|---|---|---|
| Hopper | 0.99 | 0.95 | 4 | 1e-4 | 1e-3 | 512 | 256 | 3 | 20 | 2000 | 50000 | 1000 | 40 |
| Walker2d | 0.99 | 0.95 | 4 | 1e-4 | 1e-3 | 512 | 256 | 3 | 20 | 2000 | 50000 | 1000 | 40 |
| HalfCheetah | 0.99 | 0.95 | 4 | 1e-4 | 1e-3 | 512 | 256 | 3 | 20 | 2000 | 50000 | 1000 | 40 |
| lift | 0.999 | 0.95 | 4 | 1e-4 | 5e-4 | 512 | 256 | 3 | 20 | 1200 | 7500 | 300 | 50 |
| can | 0.999 | 0.95 | 4 | 1e-4 | 5e-4 | 512 | 256 | 3 | 20 | 1200 | 7500 | 300 | 50 |
| square | 0.999 | 0.95 | 4 | 1e-4 | 5e-4 | 512 | 256 | 3 | 20 | 1600 | 10000 | 400 | 50 |
| transport | 0.999 | 0.95 | 8 | 1e-4 | 5e-4 | 512 | 256 | 3 | 20 | 3200 | 10000 | 800 | 50 |

Table 5: Hyperparameter settings for Baselines other than BDPO and DAC in robot control tasks. Experiment is executed using DPPO (Ren et al., 2024) implementation and hyperparameters. Batch size for all baselines other than DPPO is 1000. For further details, please refer to DPPO paper (Ren et al., 2024).

| Task | Rollout Samples | Rollout Interval | Warm Up | Envs | Critic Rho | Critic Samples | Replay Buffer Size |
|---|---|---|---|---|---|---|---|
| Kitchen-DAC | 10 | 5000 | - | 50 | 0 | 10 | 1e6 |
| Kitchen-BDPO | 10 | 5000 | 50000 | 50 | 0 | 10 | 1e6 |
| Robomimic-DAC | 1 | 2000 | - | - | - | 10 | 1e6 |
| Robomimic-BDPO | 1 | 2000 | 20000 | 50 | 0 | 10 | 1e6 |

Table 6: Hyperparameters for DAC and BDPO on Kitchen and Robomimic tasks. Other hyperparameters and implementation is from Gao & Cao (2025). Here we use 1 sample for rollout because we find it difficult to learn an accurate Q-estimation, which leads to severely degraded policy quality when number of samples is greater than 1.

| Task | $\gamma$ | $\lambda$ | Action Chunk | Actor LR | Critic LR | Actor MLP Size | Critic MLP Size | Actor MLP Layers | $\eta$ | Initial Noise Log Std | $1/T$ | Denoising Step | Clone Epochs | Clone LR |
|---|---|---|---|---|---|---|---|---|---|---|---|---|---|---|
| 3 vs 1 with Keeper | 0.99 | 0.95 | 1 | 5e-4 | 5e-4 | 256 | 256 | - | - | - | 20 | 5 | 8 | 1e-3 |
| Corner | 0.99 | 0.95 | 1 | 5e-4 | 5e-4 | 256 | 256 | - | - | - | 20 | 5 | 8 | 1e-3 |
| Counterattack Hard | 0.99 | 0.95 | 1 | 5e-4 | 5e-4 | 256 | 256 | - | - | - | 20 | 5 | 8 | 1e-3 |

Table 7: Hyperparameter of MLP on football. Experiment is run on MAPPO codebase and MLP architecture remains same as MAPPO, and does not use residual connection, thus rendering parameter MLP layers unusable.

| Task | $\gamma$ | $\lambda$ | Action Chunk | Actor LR | Critic LR | Actor MLP Size | Critic MLP Size | Actor MLP Layers | $\eta$ | Initial Log Std | $1/T$ | Denoising Step |
|---|---|---|---|---|---|---|---|---|---|---|---|---|
| Walker2d-NCDPO | 0.995 | 0.985 | 1 | 1e-4 | 1e-3 | 256 | 256 | 3 | 0 | -0.8 | - | 5 |
| HalfCheetah-NCDPO | 0.99 | 0.985 | 1 | 1e-4 | 1e-3 | 256 | 256 | 3 | 0 | -0.8 | - | 5 |
| Walker2d-MLP+PPO | 0.995 | 0.985 | 1 | 1e-4 | 1e-3 | 256 | 256 | 3 | - | -0.8 | - | - |
| HalfCheetah-MLP+PPO | 0.99 | 0.985 | 1 | 1e-4 | 1e-3 | 256 | 256 | 3 | - | -0.8 | - | - |
| Walker2d-DPPO | 0.99 | 0.985 | 1 | 1e-4 | 1e-3 | 512 | 256 | 3 | - | - | - | 10 |
| HalfCheetah-DPPO | 0.99 | 0.985 | 1 | 1e-4 | 1e-3 | 512 | 256 | 3 | - | - | - | 10 |

Table 8: Hyperparameters for training from scratch. In this experiment, MLP+PPO has exatcly the same architecture with MLP in diffusion's denoising process. Numbers of mini-batches and parallel environments are the same as Table 4.

| Task | $\gamma$ | $\lambda$ | Action Chunk | Actor LR | Critic LR | Actor MLP Size | Critic MLP Size | Actor MLP Layers | $\eta$ | Initial Log Std | $1/T$ | Denoising Step | Clone Epochs | Clone LR |
|---|---|---|---|---|---|---|---|---|---|---|---|---|---|---|
| Hopper | 0.995 | 0.985 | 4 | 3e-5 | 1e-3 | 1024 | 256 | 7 | 0 | -2 | - | 5/10/20 | 8 | 1e-3 |
| Walker2d | 0.995 | 0.985 | 4 | 3e-5 | 1e-3 | 1024 | 256 | 7 | 0 | -2 | - | 5/10/20 | 8 | 1e-3 |
| HalfCheetah | 0.99 | 0.985 | 4 | 3e-5 | 1e-3 | 1024 | 256 | 7 | 0 | -2 | - | 5/10/20 | 8 | 1e-3 |
| square | 0.995 | 0.985 | 4 | 3e-5 | 5e-4 | 1024 | 256 | 7 | 0 | -2.3 | - | 5/20 | 8 | 5e-4 |
| transport | 0.99 | 0.985 | 8 | 3e-5 | 5e-4 | 1024 | 256 | 7 | 0 | -2.3 | - | 5/20 | 8 | 5e-4 |

Table 9: Hyperparameter of Ablation on Denoising Steps.

| Task | $\gamma$ | $\lambda$ | Action Chunk | Actor LR | Critic LR | Actor MLP Size | Critic MLP Size | Actor MLP Layers | $\eta$ | Initial Log Std | $1/T$ | Denoising Step | Clone Epochs | Clone LR |
|---|---|---|---|---|---|---|---|---|---|---|---|---|---|---|
| lfit | 0.999 | 0.99 | 4 | 3e-5 | 1e-3 | 1024 | 256 | 7 | 0 | -2.3/-2.7 | - | 5 | 2 | 1e-3 |
| can | 0.999 | 0.99 | 4 | 3e-5 | 1e-3 | 1024 | 256 | 7 | 0 | -2.3/-2.7 | - | 5 | 60 | 3e-4 |
| square | 0.999 | 0.99 | 4 | 3e-5 | 1e-3 | 1024 | 256 | 7 | 0 | -2.3/-2.7 | - | 5 | 200 | 2e-4/5e-4 |
| transport | 0.999 | 0.99 | 8 | 3e-5 | 1e-3 | 1024 | 256 | 7 | 0 | -2.3/-2.7 | - | 5 | 321 | 8e-4/5e-4 |

Table 10: Hyperparameter of Ablation on Initial Noise.

| Task | $\gamma$ | $\lambda$ | Action Chunk | Actor LR | Critic LR | Actor MLP Size | Critic MLP Size | Actor MLP Layers | $\eta$ | Initial Log Std | $1/T$ | Denoising Step | Clone Epochs | Clone LR |
|---|---|---|---|---|---|---|---|---|---|---|---|---|---|---|
| 3 vs 1 with Keeper | 0.99 | 0.95 | 1 | 3e-5 | 1e-3 | 1024 | 256 | 7 | 0.03/0.1/0.3/1 | - | 20 | 5 | 8 | 1e-3 |
| Corner | 0.99 | 0.95 | 1 | 3e-5 | 1e-3 | 1024 | 256 | 7 | 0.03/0.1/0.3/1 | - | 20 | 5 | 8 | 1e-3 |
| Counterattack Hard | 0.99 | 0.95 | 1 | 3e-5 | 1e-3 | 1024 | 256 | 7 | 0.03/0.1/0.3/1 | - | 20 | 5 | 8 | 1e-3 |

Table 11: Hyperparameter of Ablation on $\eta$. $\eta = 1$ indicates using original scheduler.

# D    COMPUTATIONAL RESOURCES

Each run could be done in 10 hours with 1 AMD Ryzen 3990X 64-Core Processor and 1 NVIDIA 3090 GPU.

# E    FURTHER STUDY ON IMPACT OF MDP LENGTHS

We further compared the performance of training from scratch with DPPO using different number of denoising steps.

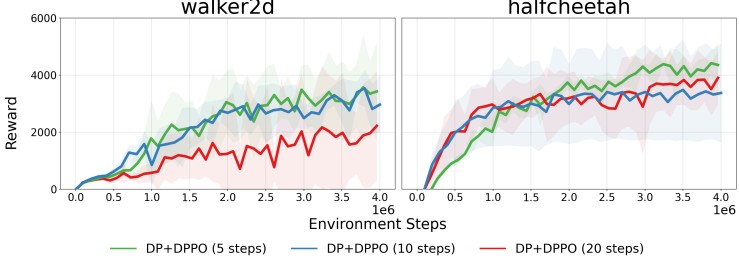

Figure 12: RL training from randomly initialized policy on Walker2D and HalfCheetah. Results are averaged over three seeds. Training curves indicate that 5 steps achieves best performance.

# F    WALL-CLOCK RESULTS

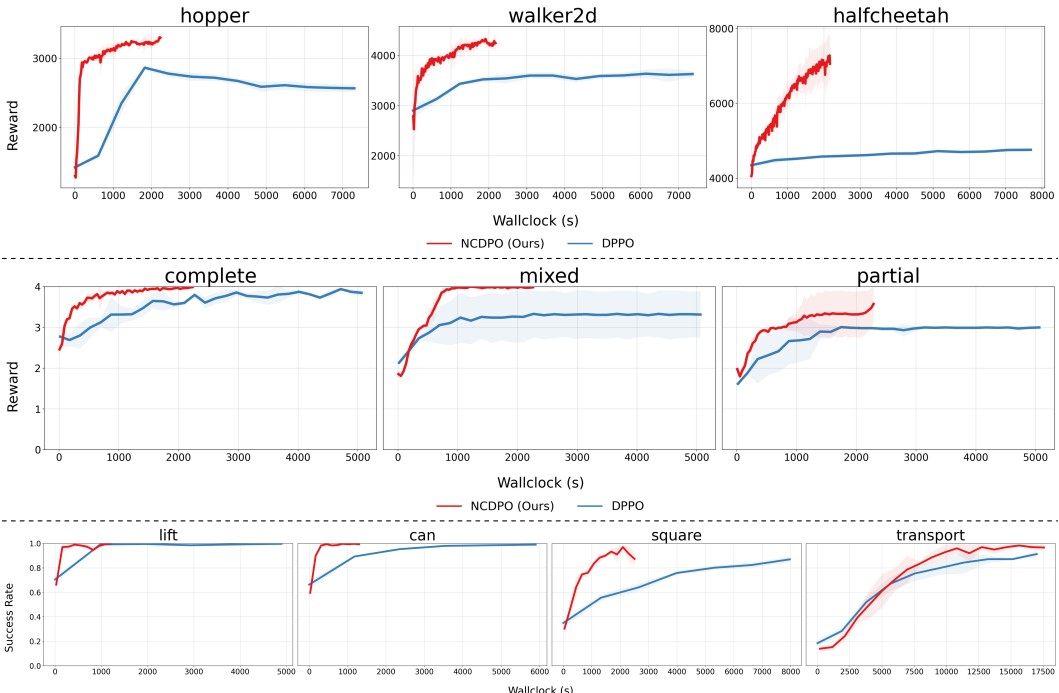

Figure 13: Wall-clock comparisons on OpenAI Gym locomotion, Franka Kitchen, and Robomimic tasks.

Theoretically, our inference speed is the same as DPPO, and the computational FLOPs required during training are also roughly equivalent. The primary difference in wall-clock speed is because DPPO uses more update epochs, leading to longer total training times.

# G    DISCUSSION ON GPU MEMORY CONSUMPTION

We report the GPU memory consumption comparison between NCDPO and DPPO in Table 12.

| Scenario | NCDPO | DPPO |
|---|---|---|
| halfcheetah | 998 | 1140 |
| walker2d | 1596 | 900 |
| hopper | 1596 | 1140 |
| | | |
| kitchen-complete | 1236 | 232 |
| kitchen-mixed | 1236 | 232 |
| kitchen-partial | 1236 | 232 |
| | | |
| lift | 1530 | 944 |
| can | 2242 | 944 |
| square | 2162 | 2078 |
| transport | 2194 | 2390 |

Table 12: Memory consumption (MB) across all control environments.

Theoretically, because we backpropagate through diffusion timesteps, we must store the activations for $T$ steps, resulting in an $O(T)$ memory requirement relative to DPPO. However, there are several ways to mitigate this overhead.

First, tuning the mini-batch size can effectively manage peak memory usage. Second, similar to standard gradient checkpointing, we can strictly store the activations of the actions in the denoising process and recompute network activations during backpropagation. *This should be very effective since action vectors are often of very low dimensions.*

Finally, we can reduce the memory complexity to $O(B \log_B T)$ by extending gradient checkpointing with a divide-and-conquer strategy, at the cost of $O(\log_B T)$ times more compute. Briefly, we define the forward and backward process of step $[l, r]$ as solve$(l, r, a_l, g_r)$, which accumulates gradients on the network parameters and returns the gradient at step $l$. Here, $a_l$ is the activation state at step $l$, and $g_r$ is the gradient at step $r$. We define split points $\mathrm{mid}_i = (l \times (B - i) + r \times i)/B$ for $i \in [1, B)$.

To execute solve$(l, r, a_l)$, we first perform a forward pass starting from the activation state on step $l$ and store the activation states $a_{\mathrm{mid}_i}$. We then enumerate $i$ from $B - 1$ down to 1 and recursively call solve$(\mathrm{mid}_i, \mathrm{mid}_{i+1}, a_{\mathrm{mid}}, g_{\mathrm{mid}+1})$ to acquire $g_{\mathrm{mid}}$. Since the recursion tree depth is $O(\log_B T)$, we need to repeat the calculation $O(\log_B T)$ times, but we only need to store $O(B)$ activations and gradients at each level. If we set $B = \sqrt{T}$, we can reduce memory complexity to $O(\sqrt{T})$ at the cost of roughly double the calculation. For a Diffusion Policy with 1000 denoising steps, we could use this method to reduce the memory consumption to only storing activation of 30 steps of actions.

## H  STABILITY OF GRADIENTS DURING BACKPROPAGATION

Empirically, we observe that the magnitude of backpropagated gradients diminishes significantly from diffusion step $t = 0$ to $t = T$. As an illustrative example, Table 13 presents a snapshot of gradient norms collected during the fine-tuning of a 20-step Diffusion Policy. The gradient magnitude decays by approximately two orders of magnitude throughout the backpropagation process.

| Step | 0 | 1 | 2 | 3 | 4 | 5 | 6 | 7 | 8 | 9 |
|---|---|---|---|---|---|---|---|---|---|---|
| Norm | 0.4489 | 0.2628 | 0.1963 | 0.1506 | 0.1172 | 0.0927 | 0.0741 | 0.0597 | 0.0487 | 0.0399 |
| Step | 10 | 11 | 12 | 13 | 14 | 15 | 16 | 17 | 18 | 19 |
| Norm | 0.0329 | 0.0273 | 0.0227 | 0.0188 | 0.0153 | 0.0123 | 0.0096 | 0.0071 | 0.0047 | 0.0023 |

Table 13: Gradient norm snapshot during backpropagation (Step 0 to 19).

We attribute this phenomenon to the scaling effect introduced during denoising, in particular the coefficient

$$\frac{\sqrt{\alpha_t}\,(1 - \bar{\alpha}_{t-1})}{1 - \bar{\alpha}_t}$$

applied to $a^k$ within the network-predicted mean $\mu_\theta(a^k, k, s)$ in the transition

$$a^{k-1} \sim \mathcal{N}\big(a^k;\, \mu_\theta(a^k, k, s),\, \sigma_k^2 I\big).$$

Although this vanishing gradient does not empirically affect the training stability of our method, we propose a simple mitigation strategy: normalizing the gradient with respect to $a_k$ to a unit norm vector. The experimental results of this modification are shown in Figure 14.

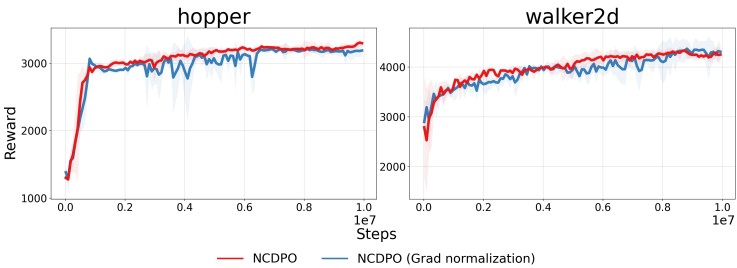

Figure 14: Performance comparison of NCDPO with and without gradient normalization.

# I USE OF LARGE LANGUAGE MODELS

Large language models are mainly used for polishing paper language, code auto-completion and minor codebase modifications.

