# OpenReview forum: "Fine-tuning Diffusion Policies with Backpropagation Through Diffusion Timesteps"
_ICLR.cc/2026/Conference — Submitted to ICLR 2026_

### Official Review · Reviewer_YqJa · 2025-10-27

**Soundness:** 3
**Presentation:** 2
**Contribution:** 2
**Rating:** 4
**Confidence:** 3

**Summary:**

The paper introduces Noise-Conditioned Diffusion Policy Optimization (NCDPO), which treats the entire diffusion denoising process as a deterministic, noise-conditioned transformation. NCDPO reframes diffusion-policy fine-tuning as a differentiable stochastic control problem. Traditional DPPO optimizes each denoising step as if it were an RL sub-policy, which is inefficient and redundant. NCDPO instead treats diffusion as a deterministic function of noise, enabling backpropagation through time without explicitly expanding the MDP. It merges the advantages of diffusion’s expressivity and PPO’s stability, while removing redundant low-level decision horizons.

**Strengths:**

- It introduces backpropagation through diffusion timesteps with fixed noise conditioning, which was previously intractable. It also avoids multi-level MDP unrolling, and a shorter effective horizon improves both convergence speed and stability.
- It works across both continuous and discrete action spaces, and scales from simple locomotion to high-dimensional vision inputs.
- All derivations align with DDPM-style parameterizations and PPO gradient logic, demonstrating how a generative diffusion model can be differentiably integrated into RL objectives, without approximations or surrogate networks.

**Weaknesses:**

- The noise-fixing assumption makes the entire process differentiable but reduces stochastic exploration. It could lead to limited diversity in gradient updates and overfitting to specific noise realizations.
- While the chain rule derivation is correct, the paper lacks a formal exposition of why this yields the same fixed-point as PPO’s stochastic gradient.
- It remains unclear whether backpropagating through deterministic denoising leads to unbiased gradient estimates compared to true stochastic diffusion.

**Questions:**

- The conceptual connection between expectation over fixed noise and policy stochasticity could be clarified.
- Is the value function $V_{\phi}(s)$ trained marginalizing over the diffusion noise or conditioned on the fixed noise sequence $z_{1:K}$?
- Empirical success is clear, but there’s no formal convergence analysis or theoretical proof that BPDT guarantees unbiased gradients.
- Memory overhead requires caching all sampled noises and intermediate activations to compute BPDT gradients, which increases memory footprint and may limit scalability for very large diffusion horizons or long-horizon tasks. Could you add a discussion on it?

---

> ### Author Response · Authors · 2025-11-24
>
> We thank the reviewer for the insightful comments and for recognizing the novelty of treating the denoising process as a differentiable control problem. We appreciate the rigorous questions regarding theoretical justification and memory scalability.
>
> **1. Fixed Noise and Policy Stochasticity (Weakness 1 & Question 1)**
> We wish to clarify the conceptual link between fixed noise and stochasticity.
> * **Reparameterization:** Our approach aligns with the standard **reparameterization trick** used in VAEs. While the noise $\epsilon$ is fixed *for the purpose of gradient calculation* (treating the transformation as deterministic given the noise), the policy itself remains stochastic because $\epsilon$ is sampled from a distribution.
> * **Fixing Noise in PPO:** In any PPO implementations with stochastic policies, the exploration noise is sampled once during the policy inference (rollout) and kept fixed throughout the computation of the policy loss and subsequent PPO updates. DPPO adheres to the same practice. Importantly, this does not diminish policy stochasticity or exploration; it merely ensures correct gradient estimation.
>
> **2. Value Function Conditioning (Question 2)**
> We confirm that the value function $V(s)$ is trained to estimate the expected return **marginalized over the diffusion noise**. It is not conditioned on the specific fixed noise $Z_{:k}$. Despite this marginalization, our gradient estimator remains unbiased.
>
> **3. Theoretical Unbiasedness (Weakness 3 & Question 3)**
> We acknowledge the need for formal justification. We have added a theoretical analysis in **Section 5.3** demonstrating that our Noise-Conditioned formulation yields an unbiased gradient estimator for the RL objective. By adhering to the reparameterization trick logic, we show that our optimization objective is equivalent to the policy gradient.
>
> **4. Memory Overhead and Scalability (Weakness 4 & Question 4)**
> We appreciate the practical concern regarding memory consumption for Backpropagation Through Time.
> * **Current Scale:** In our reported experiments, memory usage was not a bottleneck due to the moderate model sizes typical in this domain.
> * **Scalability Solutions:** We have added a detailed discussion in **Appendix G** outlining strategies for scaling to larger models or longer horizons:
>     1.  **Gradient Accumulation:** Reducing mini-batch sizes.
>     2.  **Activation Checkpointing:** Storing only the inputs to each denoising step and recomputing intermediate activations during the backward pass. This should be very effective since action vectors are often of very low dimensions.
>     3.  **Step-wise Checkpointing:** This further reduces memory complexity of storing low-dimensional action vectors from $O(T)$ to $O(\sqrt{T})$, making the method scalable to long-horizon diffusion models at the cost of twice the computation.

---

> > ### Comment · Reviewer_YqJa · 2025-11-27
> > **Response to Submission6567 Authors**
> >
> > Thank the authors for the response. I have no major concerns and will raise my recommendation to weak accept. The main paper is much clearer now. I would love to discuss with the other reviewers.

---

> ### Author Response · Authors · 2025-11-27
>
> Dear Reviewer YqJa:
>
> Thank you for taking the time to revisit our submission and adjust your score. We sincerely appreciate the effort you invested in reading our rebuttal and reevaluating our work.

---

### Official Review · Reviewer_HAzZ · 2025-10-31

**Soundness:** 3
**Presentation:** 3
**Contribution:** 2
**Rating:** 4
**Confidence:** 4

**Summary:**

The core contribution of this work is a reformulation of diffusion timesteps as noise-conditioned deterministic transformations, which unlocks efficient gradient computation and full backpropagation through the entire sampling process. This approach achieves sample efficiency comparable to established MLP-PPO methods while demonstrating significantly enhanced robustness and performance, as established through comprehensive evaluations in diverse RL settings such as robotic control and multi-agent games.

**Strengths:**

- Broad, diverse evaluation across continuous-control locomotion, multi-agent discrete environments, and manipulation (Robomimic), demonstrating robustness and applicability across domains
- Consistent gains across datasets of varying quality (medium-replay vs medium-expert), suggesting stability to data quality and strong offline-to-online transfer

**Weaknesses:**

- **Exploration and stability analysis, along with real-world validation, remain underdeveloped.** While DPPO is included as a key baseline, the study lacks visual diagnostics for exploration and stability—such as those presented in the original DPPO paper—and does not incorporate real-robot deployment to substantiate its practical applicability.

- **The multi-agent evaluation on Google Research Football is incomplete relative to the paper’s stated focus.** Given the core contribution revolves around RL optimization for diffusion policies, the GRF experiments should prioritize head-to-head comparisons against relevant diffusion-based baselines. Although several diffusion-RL methods are listed in the related work, they are not sufficiently included as comparators in the experiments.

- **Evidence for wall-clock training performance is incomplete.** While the paper claims strong performance and high sample efficiency, it does not provide  wall-clock learning curves, or time-to-threshold analyses, which are essential to rigorously support the efficiency and convergence claims.

**Questions:**

- Exploration/stability and hardware: Provide DPPO-like exploration and stability visualizations (state coverage, entropy, return variance); include at least one high-fidelity sim or real-robot deployment; report inference latency as denoising steps increase.
- Diffusion-based comparisons in GRF: Incorporate representative diffusion-RL baselines from the related work (e.g., diffusion+Q-learning, critic-guided denoising variants) under matched pretraining, data budgets, and compute to align evaluation with the paper’s stated focus

---

> ### Author Response · Authors · 2025-11-24
>
> We thank the reviewer for the positive assessment of our work's soundness and presentation, particularly regarding our evaluation across diverse domains. We value the feedback on validation and baselines and provide our responses below.
>
> **1. Exploration, Stability, and Hardware Deployment (Weaknesses 1 & Question 1)**
> * **Exploration:** We clarify that both NCDPO and the baselines utilize the same underlying diffusion architecture and pretraining datasset. Consequently, the exploration mechanics which is largely driven by the stochasticity of the diffusion noise remain structurally similar. We have also added a formal proof of the unbiasedness of our method in Section 5.3, and hence even if our formulation treats the optimization of diffusion policy as deterministic, we could still exploit the exploration from diffusion model itself.
> * **Stability:** Our empirical results (see Figure [3,4,5]) confirm the stability of NCDPO. The training curves show low-variance, monotonic improvement, avoiding the instability and performance degradation observed in some of the baseline methods.
> * **Hardware Deployment:** Regarding real-world validation, we respectfully note that standard sim-to-real transfer in DPPO relies on reconstructing states from visual observations before feeding into the policy. Therefore, the transfer success is dominated by the perception/state-reconstruction accuracy, which is orthogonal to our core contribution of improved policy optimization.
>
> **2. Google Research Football and Discrete Space (Weakness 2 & Question 3)**
> We included the Google Research Football (GRF) benchmark specifically to highlight a unique capability of NCDPO: **compatibility with discrete action spaces**.
> * **Baseline constraints:** Traditional diffusion-based RL algorithms are inherently designed for continuous spaces. To our knowledge, no other diffusion online fine-tuning method currently supports discrete actions. Therefore, we could not include diffusion+Q-learning or similar baselines for this specific task, as they are not trivially adaptable to the discrete setting without significant architectural changes that would exceed the scope of a baseline comparison.
>
> **3. Wall-Clock Performance (Weakness 3)**
> We agree that computational efficiency is a critical metric. We have added a detailed analysis of wall-clock performance in **Appendix F**.
> * **Inference Speed:** Theoretically, inference latency is identical to DPPO as we utilize the same model architecture.
> * **Training Speed:** While our method requires storing gradients over intermediate steps, the FLOPs for the backward pass are roughly same with DPPO style backpropagation. However, we find that under current DPPO paper's hyperparameter, it requires more PPO update epochs and mini-batch updates, which leads to slower training speed.

---

> ### Author Response · Authors · 2025-11-27
>
> Dear Reviewer,
>
> Thank you again for your valuable comments and suggestions. Your feedback has played a crucial role in enhancing both the quality and clarity of our paper.
>
> While the discussion period is going to end in 7 days, we have not yet received the anticipated further responses. Your insights and suggestions are not only highly appreciated but also integral to our process, and we stand ready to make any necessary improvements to the paper. If you have any additional questions or require further clarification, please do not hesitate to reach out.

---

### Official Review · Reviewer_gN21 · 2025-11-01

**Soundness:** 2
**Presentation:** 2
**Contribution:** 2
**Rating:** 4
**Confidence:** 4

**Summary:**

The paper presents the **NCDPO** algorithm, a novel diffusion policy fine-tuning framework that reformulates the **Diffusion Policy** as a **noise-conditioned deterministic policy**. NCDPO treats each denoising step as a **differential transformation** and fine-tunes the **fully denoised interactive actions**. Extensive experiments on **OpenAI Gym**, **Franka Kitchen**, **RoboMimic**, and **Google Research Football** demonstrate that **NCDPO** achieves superior performance compared to existing diffusion policy fine-tuning methods.

**Strengths:**

1. The paper tackles a **highly important and timely research question**, particularly as diffusion models are becoming increasingly influential in the domains of **imitation learning**, **reinforcement learning**, and **Vision-Language-Action (VLA)** modeling.

2. The paper is **well-written** and **clearly organized**, effectively presenting its motivation, methodology, and experimental results.

3. The proposed **NCDPO** algorithm demonstrates **strong performance** compared to other **Diffusion Policy online fine-tuning** methods.

**Weaknesses:**

### Major Weaknesses:

1. I am somewhat skeptical about the authors’ claim that the **low sample efficiency of DPPO** stems from a **lengthened MDP horizon**. More evidence is needed to substantiate this argument, as it is closely tied to the motivation of the paper. If a longer MDP horizon truly causes **DPPO** to be sample inefficient, then **fine-tuning DDIM** with fewer diffusion denoising steps, or fine-tuning only the final few denoising steps of DDPM, should yield sample efficiency improvements compared to fine-tuning all denoising steps with **DDPM**. Additionally, I find the experiments in **Section 4** somewhat unclear in purpose. The results appear self-evident, since (1) **DPPO** is not designed to train a randomly initialized Diffusion Policy from scratch, and (2) greater representational capacity does not necessarily translate to better performance, particularly in tasks where such capacity is not required.

2. The authors should clarify the **core contribution** of the proposed **NCDPO** algorithm. At present, it appears conceptually similar to directly fine-tuning a **Diffusion Policy** using **PPO** in a standard manner. It is generally acknowledged that **backpropagation through multiple denoising steps** can lead to unstable gradients. Therefore, the authors are encouraged to explain in greater detail how **reformulating Diffusion Policy as a deterministic policy** effectively mitigates this issue.

3. In the **Google Research Football** benchmark, the authors compare **NCDPO** with **MLP + MAPPO**. It is unclear what this comparison is intended to demonstrate. At a minimum, the authors should include **Diffusion Policy online fine-tuning methods** as baselines to provide a more meaningful and fair evaluation.

4. In several tasks, including **OpenAI Gym** and **Franka Kitchen**, **NCDPO** achieves higher final performance than **DPPO**. The authors are strongly encouraged to discuss why **NCDPO** not only improves **sample efficiency** but also enhances **final performance** in certain cases.

### Minor Weakness:

1. It is observed that **NCDPO** achieves significantly higher **sample efficiency** in the **RoboMimic Transport (visual)** setting compared to the **state-based** setting. The authors are encouraged to provide some discussion or analysis to explain this discrepancy.

2. The authors are encouraged to discuss recently proposed **Diffusion Policy fine-tuning methods** that do not require updating the parameters of the diffusion model itself. Including comparisons with such approaches would further strengthen the paper and is recommended for future work.

Wagenmaker, Andrew, et al. "Steering Your Diffusion Policy with Latent Space Reinforcement Learning." arXiv preprint arXiv:2506.15799 (2025).

Yuan, Xiu, et al. "Policy decorator: Model-agnostic online refinement for large policy model." arXiv preprint arXiv:2412.13630 (2024).

Ankile, Lars, et al. "From imitation to refinement-residual rl for precise assembly." 2025 IEEE International Conference on Robotics and Automation (ICRA). IEEE, 2025.

**I am more than willing to raise my scores if the authors adequately address my concerns.**

**Questions:**

1. **(Related to Major Weakness 1)** Why do the authors believe that a **lengthened MDP horizon** is the main cause of the **low sample efficiency** observed in **DPPO**? What are the key insights that the experiments in **Section 4** are intended to convey?

2. **(Related to Major Weakness 2)** What are the **main contributions and novelties** of this paper? How does **NCDPO** effectively mitigate the **unstable gradients** that arise from **backpropagation through multiple denoising steps**?

3. **(Related to Major Weakness 3)** What is the intended takeaway from the **Google Research Football** experiments? What do the authors aim to demonstrate with this comparison?

4. **(Related to Major Weakness 4)** Why does **NCDPO** not only improve **sample efficiency** but also achieve **higher final performance** compared to **DPPO**?

---

> ### Author Response · Authors · 2025-11-24
>
> We sincerely thank the reviewer for the detailed evaluation and constructive feedback. We appreciate the recognition of our work’s timeliness, clarity, and performance. Below, we address the specific concerns raised.
>
> **1. Impact of MDP Horizon on Sample Efficiency**
> We appreciate the reviewer's skepticism regarding the "lengthened MDP" hypothesis. Our hypothesis is based on the observation that DPPO has much lower sample efficiency than standard MLP policy and we attribute this to longer horizon length. To validate this, the experiments in Section 4 were designed to isolate the horizon factor by removing Behavior Cloning and leaving out the RL components.
>
> Furthermore, as suggested, **we have added Appendix E**, which includes further results of training DPPO from scratch with different number of denoising steps. We observed that a denoising horizon of 5 steps yields better results than 10 or 20 steps, supporting the claim that longer horizons hinder optimization in this context.
>
> Regarding DDIM: while DPPO already utilize DDIM for visual tasks, it still underperforms.
>
> **2. Core Contribution and Methodological Distinctness**
> We apologize for any ambiguity regarding our contributions. The core novelty of NCDPO is the introduction of **Backpropagation Through Diffusion Timesteps (BPDT)** for PPO-based fine-tuning, which exploits the differentiability of the denoising process, and helps to improve the sample efficiency. While the concept may appear straightforward, to the best of our knowledge, no prior work has proposed this approach.
> * **Theoretical Analysis:** We have added a theoretical analysis in **Section 5.3** proving the unbiasedness of our gradient estimator, and showed the connection between our method and the reparametrization trick adopted in methods like VAE.
> * **Generalization:** We demonstrate that our method is applicable to any policy parameterization where the final output is parameterized by $a^0$ (e.g., Gaussian or Softmax). To our knowledge, this is the first method capable of online fine-tuning for diffusion policies in **discrete action spaces**.
> * **Gradient Stability:** We have added a discussion on gradient stability in **Appendix H**, and proposed a simple solution to mitigate the gradient vanishing issue.
>
> **3. Google Research Football and Baselines**
> The primary intent of the GRF experiments is to demonstrate the versatility of NCDPO in handling **discrete action spaces**, a domain where traditional diffusion policies struggle due to their continuous nature.
> * **Reason for Baseline Choice:** We compared against MLP-MAPPO because, to the best of our knowledge, there are no existing diffusion-based online fine-tuning methods that support discrete action spaces. We believed comparing against MAPPO which is a strong baseline in discrete action space is a proper approach in the absence of direct diffusion competitors.
>
> **4. Improvements in Sample Efficiency and Final Performance**
> We appreciate the opportunity to clarify why NCDPO improves both metrics:
> * **Sample Efficiency:** This stems from the backpropagation through diffusion timesteps technique that effectively exploits the differentiability of the denoising process. The intuition is that this is similar to how differentiable RL is more sample efficient than policy gradient methods.
> * **Final Performance:** We hypothesize that the noise in DPPO's optimization process can destabilize the pre-trained weights, "forgetting" the cloned behavior from expert demonstrations. In complex environments like Kitchen, discovering the goal via exploration alone is intractable; preserving the demonstrated behavior is crucial. NCDPO’s stable updates better preserve these priors, leading to higher performance.
>
> **5. Visual vs. State:** We acknowledge the discrepancy in Robomimic Transport. We believe this is because in this task, visual input gives more structured information than state input. This can be validated by the fact that DPPO also achieves slightly better sample efficiency on visual task than state task.

---

> ### Author Response · Authors · 2025-11-27
>
> Dear Reviewer,
>
> Thank you again for your valuable comments and suggestions. Your feedback has played a crucial role in enhancing both the quality and clarity of our paper.
>
> While the discussion period is going to end in 7 days, we have not yet received the anticipated further responses. Your insights and suggestions are not only highly appreciated but also integral to our process, and we stand ready to make any necessary improvements to the paper. If you have any additional questions or require further clarification, please do not hesitate to reach out.

---

### Author Response · Authors · 2025-11-24
**Summary of paper updates**

We sincerely thank all reviewers for their thorough evaluations and constructive feedback. In response, we have revised the paper to address the raised concerns. Below is a summary of the key updates:

**Summary of Updates:**

1. **Section 5.3:** We have added a formal proof demonstrating the unbiasedness of the gradient estimator used in our method.
2. **Appendix E:** We included supplementary experiments supporting the hypothesis that the increased MDP horizon is a primary factor contributing to the reduced sample efficiency observed in DPPO.
3. **Appendix F:** We added a comparative analysis of the wall-clock training performance between NCDPO and DPPO.
4. **Appendix G:** We addressed concerns regarding memory consumption and outlined strategies to ensure scalability for larger models.
5. **Appendix H:** We added a discussion on gradient stability and proposed a mitigation strategy for the gradient-vanishing issue.

We hope these updates address the reviewers’ concerns.

---

### Author Response · Authors · 2025-12-01
**Note on post-rebuttal updates prior to the OpenReview incident**

We sincerely thank the Area Chair for their additional efforts and the reviewers for their thorough evaluations and constructive feedback. Below, we provide a consolidated overview of the concerns raised, the corresponding revisions made to the manuscript, and the current status of each discussion.

## Reviewer gN21

**Key Concerns:** The reviewer questioned: (1) the validity of our hypothesis that the lengthened MDP horizon leads to reduced sample efficiency in DPPO; (2) the distinctiveness of our contribution relative to standard PPO fine-tuning; and (3) the choice of baselines for the Google Research Football (GRF) benchmark.

**Our Response:**

1. We added **Appendix E**, which includes new experiments designed to isolate and empirically validate the impact of MDP horizon length on optimization efficiency.
2. We clarified that our main contribution is **Backpropagation Through Diffusion Timesteps (BPDT)**—a substantive departure from existing diffusion policy optimization techniques that enables optimization in both continuous and discrete action spaces. We also added a theoretical analysis of unbiasedness and elaborated on the connection to the reparameterization trick.
3. We added **Appendix H**, detailing the gradient stability properties of BPDT.
4. We justified our baseline selection by noting the absence of diffusion-based fine-tuning methods for discrete action spaces in existing literature.

## Reviewer HAzZ

**Key Concerns:** The reviewer requested: (1) additional visualizations illustrating exploration behavior and stability; (2) hardware deployment results; (3) wall-clock performance comparisons; and (4) inclusion of diffusion-RL baselines tailored to discrete tasks.

**Our Response:**

1. We clarified that exploration characteristics are inherently similar to baselines due to the shared diffusion architecture, and noted that the strong stability of NCDPO is evident from the training curves.
2. We explained that hardware deployment is orthogonal to the scientific scope of this work.
3. We added **Appendix F**, providing a comparative wall-clock performance analysis demonstrating NCDPO’s efficiency.
4. We explained the architectural constraints that prevent trivial adaptations of continuous diffusion-RL baselines to the discrete GRF task.

## Reviewer YqJa

**Key Concerns:** The reviewer asked for: (1) a theoretical proof regarding the unbiasedness of the NCDPO gradient estimator; (2) clarification of how the fixed-noise assumption during PPO updates relates to policy stochasticity; and (3) analysis of the memory requirements associated with storing the denoising chain.

**Our Response:**

1. We added **Section 5.3**, which now contains a formal proof of the unbiasedness of our gradient estimator.
2. We clarified the distinction between deterministic gradient computation via reparameterization and the stochasticity inherent in policy rollouts.
3. We added **Appendix G**, which discusses memory considerations and outlines scalable mitigation strategies such as gradient checkpointing.

After reviewing our responses, the reviewer confirmed that all concerns were fully addressed and raised the score from **4 to 6**. We note that this discussion and the score update were completed prior to the recent OpenReview incident.


## Summary of Core Contributions

Before concluding, we would like to respectfully reiterate the central contributions of our work, which we believe address the reviewers’ concerns and clarify the significance of our approach:

1. **Backpropagation Through Diffusion Timesteps for PPO Fine-Tuning:**
   We introduce a novel technique that leverages the differentiability of the denoising process to propagate gradients through diffusion timesteps during PPO-based fine-tuning. This significantly improves sample efficiency and, to the best of our knowledge, is the first method to apply such backpropagation within PPO for diffusion-policy fine-tuning. We also show that using this method, NCDPO could be conveniently applied to both continuous and discrete action spaces.

2. **Theoretical Guarantee of the NCDPO Gradient Estimator:**
   We provide a formal proof of the unbiasedness of the NCDPO gradient estimator and establish its connection to the classical reparameterization trick, offering a principled justification for our method.

3. **Extensive Empirical Evaluation:**
   Through comprehensive experiments on both discrete and continuous control benchmarks—including both state-based and vision-based tasks—we demonstrate that NCDPO achieves strong effectiveness, superior efficiency, and consistent improvements over prior diffusion-policy fine-tuning approaches. We further show that our method is robust to variations in the number of denoising steps.

---

### Meta-Review · Area_Chair_V4XT · 2026-01-01

**Summary:**

This paper presents a new method NCDPO, an algorithm designed to improve the sample efficiency of Diffusion Policies via a deterministic reformulation. The reviewers thought the comparisons in GRF experiment is not complete, and expressed that more theoretical analysis is needed. I think most concerns have been addressed, while some limitations remain, specifically regarding the lack of real-world validation and clarification of the core contribution . I think although this paper provides a meaningful contribution to the field, it can be still improved in a certain way. Provided that there are two reviewers keeping the score 4, I lean toward rejection.

**Reviewer Concerns:**

Reviewer gN21: This reviewer showed skepticism regarding the hypothesis that lengthened MDP horizons cause DPPO’s sample inefficiency, requested a clearer distinction between NCDPO’s core contributions and standard PPO, challenged the adequacy of the GRF task baselines, and asked for an explanation of why NCDPO improves both sample efficiency and peak performance. I think the concerns have been mostly addressed, while the core contribution still needs more clarification.

Reviewer HAzZ: This reviewer raised concerns about underdevelopement of exploration and stability analysis, and real-world validation. In addition, this reviewer states that the experiment GRF is incomplete, and asked for including wall-clock performance comparisons. Again, most of the concerns have been addressed from my point of view, while I think a real-robot deployment would have significantly strengthened the practical claims of the paper.

Reviewer YqJa: This reviewer noted that while NCDPO is empirically successful, more theoretical analysis is needed. Additionally, the reviewer asked for explanation of using fixed noise, and analysis of the memory requirements. The authors provided sufficient clarification during the rebuttal, leading the reviewer to increase the score from 4 to 6.

**Reviewer Scores:**

Reviewer YqJa increased the score from 4 to 6.

---

### Decision · Program_Chairs · 2026-01-26

Reject